

# An update and beyond: key landscapes for conservation land cover and change monitoring, thematic and validation datasets for the African, Caribbean and Pacific region

Zoltan Szantoi[1,2], Andreas Brink[1], Andrea Lupi[1]

[1]European Commission, Joint Research Centre, 21027 Ispra, Italy
[2]Department of Geography and Environmental Studies, Stellenbosch University, Stellenbosch 7602, South Africa

Correspondence to: Zoltan Szantoi (zoltan.szantoi@remote-sensing-biodiversity.org)

**Abstract.** Natural resources are increasingly being threatened in the world. Threats to biodiversity and human well-being pose enormous challenges to many vulnerable areas. Effective monitoring and protection of sites with strategic conservation importance require timely monitoring with special focus on certain land cover classes which are especially vulnerable. Larger

ecological zones and wildlife corridors warrant monitoring as well, as these areas have an even higher degree of pressure and habitat loss as they are not "protected" compared to Protected Areas (i.e. National Parks, etc.). To address such a need, a satellite-imagery-based monitoring workflow to cover at-risk areas was developed. During the program's first phase, a total of 560 442km$^2$ area in sub-Saharan Africa was covered. In this update we remapped some of the areas with the latest satellite images available, and in addition we added some new areas to be mapped. Thus, in this version we updated and mapped an

additional 852 025km$^2$ in the Caribbean, African and Pacific regions with up to 32 land cover classes. Medium to high spatial resolution satellite imagery was used to generate dense time series data from which the thematic land cover maps were derived. Each map and change map were fully verified and validated by an independent team to achieve our strict data quality requirements. The independent validation datasets for each Key Landscape for Conservation (KLC) are also described and presented here (all presented datasets are available at https://doi.org/10.5281/zenodo.4621375, Szantoi et al., 2021).

**1   Introduction**

Key landscapes for conservation (MacKinnon et al., 2015) (KLCs) are defined as areas vast enough to sustain large wild animals (e.g., "big-five" game) within functioning biomes that face pressure from various external factors such as poaching, agriculture expansion, and urbanization. Land use changes cause loss in both flora and fauna by altering wild animal movements that can lead to decreases in population size over time (Di Minin et al., 2016; van der Meer, 2018). The livelihood

of people and wildlife in the Organization of African, Caribbean and Pacific States (OACPS) that depend on natural resources



faces increasing pressure from resource consumption by the regions' growing population, for example Africa set to reach 2 billion by 2040 (MacKinnon et al., 2015; Di Minin et al., 2016). The representative location types, often transboundary, of the KLCs uniquely positions them as benchmarks for their natural resource management to generate steady income for the local residents while protecting their wildlife (MacKinnon et al., 2015). Benchmarking activities of this kind require highly accurate

thematic land cover change (LCC) map products. Although LCC maps exist for many areas within the regions, the majority of products only cover protected areas with some buffer zones (Szantoi et al., 2016). However, continental and global mapping efforts reported thematic accuracies for such land cover maps between 67% and 81 %, with lower class accuracies reported in many cases (Mora et al., 2014). Differences in legends and unstandardized methods make these cases difficult to use for monitoring, modeling, or change detection studies. In order to use various land cover (LC) and LCC products together (i.e.,

modeling, policy making), land cover class definitions should be standardized to avoid discrepancies in thematic class understanding. Not all users (international organizations, national governments, civil societies, researchers) have the capabilities to readjust such maps (Saah et al., 2020). To accommodate such diverse user profiles, a common processing scheme is employed and the resulting datasets can be utilized through various platforms and systems. This work adopted the Land Cover Classification Scheme of the Food and Agriculture Organization (FAO LCCS; Di Gregorio, 2005), an

internationally approved ISO standard. The presented datasets in this paper are produced within the Copernicus High-Resolution Hot Spot Monitoring (C-HSM) activity of the Copernicus Global Land Service.

All C-HSM products feature the same thematic land cover legend and geometric accuracy and were processed and validated following the same methodology. All products, including the C-HSM data, are free and open to any user with guaranteed long-term maintenance and availability under the Copernicus license.

Copernicus serves as an operational program where data production takes place on a continuous basis. This paper presents an update of four previously published (Szantoi et al., 2020b) land cover/change maps (Greater Virunga, Salonga, Upemba and Yangambi KLCs) covering 160 281km$^2$ of terrestrial land area in sub-Saharan Africa (SSA) and six additional KLCs covering 691 744km$^2$ in the OACPS regions. The datasets are based on freely available medium spatial-resolution data (Copernicus Sentinel-2 and USGS Landsat 5 and 8) a part of one area (Timor Leste) where we used high-spatial resolution data (SPOT4,

5, 6). Each of the KLCs were individually validated for both present and change dates. The developed processing chain always consists of preliminary data assessment for availability, pre- and post-processing, and fully independent quality verification and validation steps. For the latter, a second dataset called validation data is presented. Several recent studies call for the sharing of product validation datasets (Fritz et al., 2017; Tsendbazar et al., 2018), especially if a collection received financial support from government grants (Szantoi et al., 2020a). Accordingly, the validation datasets (LC–LCC) associated with each

of the KLCs are also shared.




## 2    Study area

The provided thematic datasets concentrate on sub-Saharan Africa with additional KLCs in the Caribbean and Pacific regions. The selection of areas was conducted based on present and future pressures envisioned and predicted by MacKinnon and colleagues (2015) and the Biodiversity and Protected Areas Management (BIOPAMA, https://biopama.org/) Programme. In this second phase (Phase 2), 10 large areas totalling 852 025km$^2$ were selected, mapped and or updated, and validated (Fig. 1). These areas cover various ecosystems and generally reside in transboundary regions (Table 1, Fig. 1).

**Figure 1 Spatial distribution of the key landscapes for conservation Phase 2 areas.**

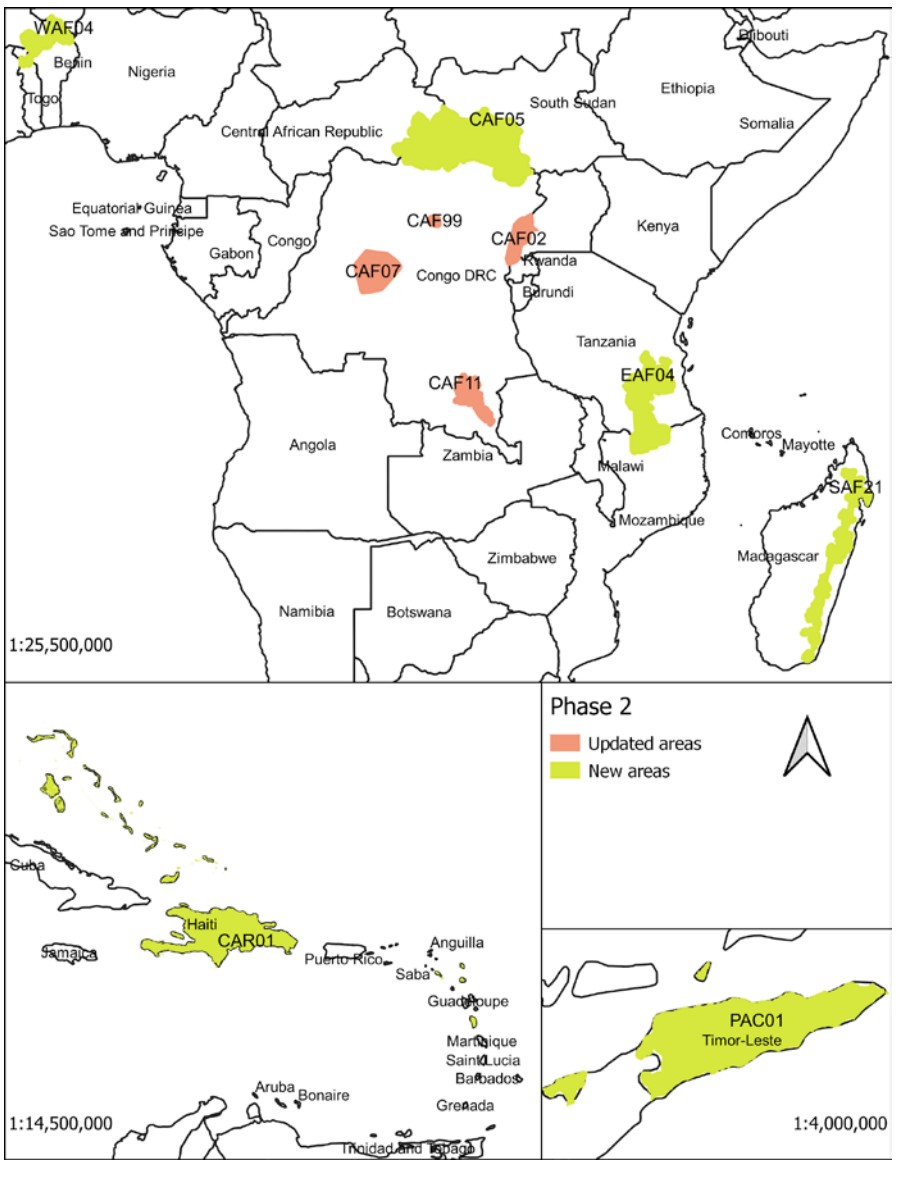





**Table 1 Mapped key landscapes for conservation within Phase 2.**

| KLC | Code | Ecoregion (Dinerstein et al., 2017) | Country | Area (km²) |
|---|---|---|---|---|
| Updated areas | | | | |
| Greater Virunga | CAF02 | Albertine Rift montane forests Victoria Basin forest–savanna | DRC, Uganda, Rwanda | 39 062 |
| Salonga | CAF07 | Central Congolian lowland forests | DRC | 66 625 |
| Upemba | CAF11 | Central Zambezian wet miombo woodlands | DRC | 47 318 |
| Yangambi | CAF99 | Northeast Congolian lowland forests | DRC | 7276 |
| New areas | | | | |
| Garamba | CAF05 | East Sudanian savanna, Northern Congolian forest-savanna mosaic, Northeastern Congolian lowland forests | DRC, Central African Republic, South Sudan | 265976 |
| Caribbean | CAR01 | Windward Islands moist forests, Bahamian-Antillean mangroves, Caribbean shrublands, Lesser Antillean dry forests, Hispaniolan moist forests, Enriquillo wetlands, Hispaniolan dry forests, Hispaniolan pine forests, Bahamian pineyards | Dominican Republic, Haiti, Bahamas, Saints Kitts and Nevis, Antigua and Barbuda, Dominica | 89883 |
| Niassa Selous | EAF04 | Zambezian flooded grasslands, Eastern Miombo woodlands, Eastern Arc forests, Northern Zanzibar-Inhambane coastal forest mosaic | Tanzania, Mozambique | 139163 |
| Timor-Leste | PAC01 | Timor and Wetar deciduous forests | Timor-Leste | 14931 |
| Madagascar | SAF21 | Madagascar lowland forests, Madagascar subhumid forests | Madagascar | 124012 |
| Wapok | WAF04 | West Sudanian savanna | Ghana, Togo, Benin, Burkina Faso, Niger | 57776 |

DRC: Democratic Republic of the Congo.





## 3 Data and method

The production workflow for the entire process is shown in Figure 2. Each stage is explained in detail in the below sections.

**Figure 2 Overall production workflow**

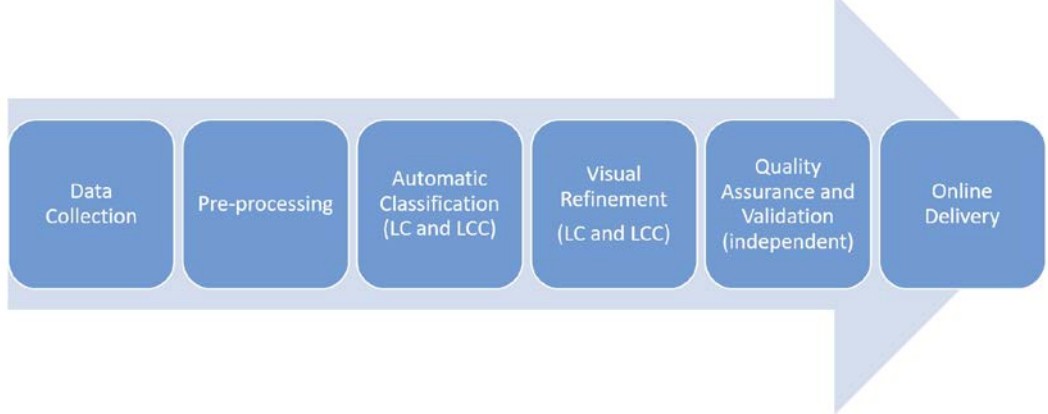

### 3.1 Data collection and mapping guidelines

Landsat TM, ETM+ and OLI at Level1TP, Sentinel-2 at Level1C, and SPOT 4, 5 and 6 at Level1-B processing level imagery were used in the production and update of the land cover and change maps. The Level1TP (Landsat), Level1C (Sentinel-2),

and Level1-B (SPOT) data were further corrected for atmospheric conditions to produce surface reflectance products for the classification phase. The atmospheric correction module was implemented based on the 6S as a direct radiative transfer model for Landsat (Masek et al., 2006) and SPOT (Haifeng et al., 2010) and using the Sen2Cor processor (v2.8) based on the ATCOR model (Richter et al., 2012). The Shuttle Radar Topography Mission (30m or 90m) Digital Elevation Model was used to estimate the target height and slope, as well as correct the surface sun incidence angles to perform an optional topographic

correction. Based on the area's meteo-climatic conditions (climate profile and precipitation patterns), season specific satellite image data were selected for each KLC (Table 1). Due to data scarcity for many areas, especially for the change maps (i.e. year 2000), imagery was collected for a target year ± 3 years. In extreme cases, (±) 5 years were allowed, or until four cloud free observations per pixel for the specified date were reached.

### 3.2 Land cover classification system

All thematic maps were produced at both *Dichotomous* and *Modular* levels within the Land Cover Classification System (LCCS) developed by the Food and Agriculture Organization of the United Nations and the United Nations Environment Programme (Di Gregorio, 2005). The LCCS (ISO 19144-2) is a comprehensive hierarchical classification system that enables comparison of land cover classes regardless of geographic location or mapping date and scale (Di Gregorio, 2005). At the *Dichotomous* level, the system distinguishes eight major LC classes. At the *Modular* level, thirty-two LC classes were used



(Table 2). For the Caribbean (CAR01), Timor-Leste (PAC01), and Madagascar (SAF21) KLCs, we included an additional land cover class not present in other KLC map products: "Not Inland Cover", due to the special location and of the mapped areas (i.e. islands), this class is not present in LCCS and we only used it for our error assessment.

**Table 2 Dichotomous and Modular thematic land cover/use classes (MCD - mapcode dichotomous level, MCM - mapcode modular level, AG - aggregated classes for land cover change accuracy estimation, see section 3.5 for additional information).**

| Dichotomous level | MCD | Modular level | MCM | AG |
|---|---|---|---|---|
| Cultivated and Managed Terrestrial Area (A11) | 3 | continuous large to medium sized field (>2 ha) of tree crop cover: plantation | 31 | 3 |
| | | continuous small sized field (<2 ha) of tree crop cover: plantation | 32 | 3 |
| | | continuous large to medium sized field (>2 ha) of tree crop cover: orchard | 33 | 3 |
| | | continuous small sized field (<2 ha) of tree crop cover: orchard | 34 | 3 |
| | | continuous large to medium sized field (>2 ha) of shrub crop | 55 | 3 |
| | | continuous small sized field (<2 ha) of shrub crop | 56 | 3 |
| | | continuous large to medium sized field (>2 ha) of herbaceous crop | 59 | 3 |
| | | continuous small sized field (<2 ha) of herbaceous crop | 60 | 3 |
| Natural and Semi-Natural Primarily Terrestrial Vegetation (A12) | 4 | continuous closed (>70-60) trees | 77 | 77 |
| | | continuous open general (70-60)-(20-10)% trees | 78 | 78 |
| | | continuous closed to open (100-40)% shrubs | 112 | 4 |
| | | continuous open (40 - (20-10)%) shrubs | 116 | 4 |
| | | continuous closed to open (100-40)% herbaceous vegetation | 148 | 4 |
| | | continuous open (40 - (20-10)%) herbaceous vegetation | 152 | 4 |
| Cultivated Aquatic or Regularly Flooded Area (A23) | 6 | continuous large to medium sized field (>2 ha) of woody crops | 155 | 6 |
| | | continuous small sized field (<2 ha) of woody crops | 156 | 6 |
| | | continuous large to medium sized field (>2 ha) of graminoid crops | 159 | 6 |



| | | | | |
|---|---|---|---|---|
| | | continuous small sized field (<2 ha) of graminoid crops | 160 | 6 |
| Natural And Semi-Natural Aquatic or Regularly Flooded Vegetation (A24) | 7 | closed (>70-60)% trees | 165 | 165 |
| | | open general (70-60)-(20-10)% trees | 166 | 165 |
| | | closed to open (100-40)% shrubs | 171 | 7 |
| | | very open (40 - (20-10)%) shrubs | 175 | 7 |
| | | closed to open (100-40)% herbaceous vegetation | 178 | 7 |
| | | very open (40 - (20-10)%) herbaceous vegetation | 182 | 7 |
| Artificial Surfaces and Associated Area (B15) | 10 | built up area | 184 | 184 |
| | | non built up area | 185 | 185 |
| Bare Area (B16) | 11 | Bare area | 11 | 11 |
| Artificial Waterbodies, Snow and Ice (B27) | 13 | artificial waterbodies (flowing) | 186 | 13 |
| | | artificial waterbodies (standing) | 187 | 13 |
| Natural Waterbodies, Snow and Ice (B28) | 14 | natural waterbodies (flowing) | 190 | 14 |
| | | natural waterbodies (standing) | 191 | 14 |
| | | snow | 192 | 14 |
| | | ice | 193 | 14 |
| Not Inland Cover | 99 | not terrestrial cover | 999 | 999 |

## 3.3 Automatic classification

Based on the pre-selected imagery data (Landsat, Sentinel-2, and SPOT), Dense Multitemporal Timeseries (DMT) based vegetation indices were generated to reduce data dimensionality and enhance the signal of the surface target. The DMT for each KLCs were based on the pre-processed and geometrically coregistered data, forming a geospatial datacube (Strobl et al., 2017). In addition, three vegetation indices were calculated to aid the separation of terrestrial vs. aquatic (NDFI), vegetated vs. barren (SAVI), and evergreen vs. deciduous vegetation areas (NBR).

The indices are (per Landsat spectral bands):

*Normalized Difference Flooding Index (NDFI)*          $NDFI = \frac{(RED-SWIR)}{(RED+SWIR)}$          *(1)*




| Soil Adjusted Vegetation Index (SAVI) | $SAVI = \frac{1.5x(NIR-RED)}{(NIR+RED+0.5)}$ | (2) |

| Normalized Burn Ratio (NBR) | $NBR = \frac{(NIR-SWIR)}{(NIR+SWIR)}$ | (3) |

All the pre-processed data (spectral bands and the DMT based indices) were fed into the Support Vector Machine supervised classification model. The Support Vector Machine classifier can handle data with high dimensionality and performs well with mapping heterogeneous areas, including vegetation community types (Szantoi et al., 2013). To produce the thematic maps, the Minimum Mapping Unit concept used by Szantoi et al. (2016) was employed. Individual pixels (with corresponding land cover class information) were assigned into objects, where the minimum size of an object was set at 3 hectares (0.03km$^2$), as a
compromise between technical feasibility (pixel size) and the general size of the observable features (various land cover classes). Still, classification errors (omission and commission of various classes) and false alarms (for land cover change) arose due to the data availability (cloud cover, no data) and the seasonal behaviour of the land cover (e.g. rapid foliage change). To correct these errors, expert human image interpretation skills and knowledge that improved the outputs from the automated process were employed.

**3.4 Land cover change detection**

Land cover change was interpreted as a categorical change in which a particular land cover was replaced by another land cover. As an example of conversion, the change of Cultivated and Managed Terrestrial Areas (A11) into a Natural and Semi-Natural Terrestrial Vegetation (A12) or a Cultivated and Managed Terrestrial Areas (A11) into Artificial Surfaces and Associated Areas (B15) can be mentioned. The basic condition for LC changes identification was the detection of changes in spectral
reflectance within specific image bands of the employed satellite imagery and in the generated indices, but such changes were further evidenced by other interpretation parameters such as shape and texture patterns. In regards to our methodology, images acquired in two or more different timeframes were used in the identification process. Furthermore, land cover changes were characterized by those changes that have longer than yearly and/or seasonal periodicity (dry/wet season). Urban sprawl, tree plantations (large or small) to replace herbaceous crops (large or small), tree covers (closed or open) or the creation of a new
water reservoir undergo long-term changes that classify as actual LCCs. In our workflow, the LCC process followed the same image pre-processing steps as the LC method, and an independent classification (similarly to the LC procedure) of the past date was performed. Finally, the LC and the LCC products were compared and change polygons (minimum of 0.5 hectare change) were extracted. As with the LC product, the visual refinement was an important step to produce accurate LCC polygons.





### 3.5 Validation dataset production

The validation datasets (Table 3, Figures 3 and 4) were individually created for each KLCs. The validation datasets (points) were generated using a stratified random sampling procedure. This assured a sufficient estimation for all land cover and land cover change classes according to their frequency of occurrence. The following formula (Gallaun et al., 2015) was used to determine the minimum number of validation points (per class per KLC):

$$n_c = \frac{p_{c(1-p_c)}}{\sigma_c^2}, c = 1, \dots, L \tag{4}$$

$n_c$ number of sampling units for class c

$p_c$ estimated error rate for class c

$\sigma_c$ accepted standard error of the error of commission for class c

$L$ number of classes

In cases where classes covered smaller areas in total, additional sampling units were allocated according to the Neyman optimal allocation in order to minimize the variance of the estimator of the overall accuracy for the total sample size [n] (Gallaun et al., 2015; Stehman, 2012):

$$n_c = \frac{nN_c\sigma_c}{\sum_{k=1}^{L} N_k\sigma_k} \tag{5}$$

$n_c$ sample size for class c

$N_c$ population size for class c

$\sigma_c$ estimated error rate for class c

$L$ number of classes

$N_k$ population size for class k

$\sigma_k$ estimated error rate for class k

At least two independent data analysts (blind and plausibility interpretation process) evaluated all accuracy points. Some points were excluded from the accuracy statistics due to an error/disagreement during the evaluation procedure (Table 3 - "Number of points LC/LCC"). The *blind* process attempt to interpret all validation points was based on available ancillary data (i.e. higher resolution imagery), without direct comparison to the generated LC/LCC maps. The *plausibility* process reviewed every point whose the blind interpretation did not match the corresponding LC/LCC value (disagreement between the LC/LCC data and the blind interpretation). After this review, the final validation reference is established.

The validation of the change maps (apart of CAF07, where we have assessed all the LCCS modular classes) aimed to assess the accuracy of the change detection. Thus, the following change categories were evaluated for those land cover changes (i.e.



the accuracy assessments were done based on the below aggregated LCCS classes) - the aggregated classes are also presented
in Table 2.

- Loss of natural vegetation - change from vegetation classes to any other class

- Gain of natural vegetation - change from any class to vegetation classes

- Woody natural vegetation (forest) cover loss - tree cover to any other class

- Woody natural vegetation (forest) cover gain - change from any class to tree cover

- Woody natural vegetation (forest) degradation - change from closed forest to open forest

- Woody natural vegetation (forest) regeneration - change from open forest to closed forest

- Cultivated and managed (cropland) extension - change from any class to cultivated classes

- Artificial surfaces (Human settlements) expansion - change from any class to built-up class

**Table 3 Validation dataset attributes**

| KLC Code | Land cover | | Land cover change | | Number of points |
|---|---|---|---|---|---|
| | Number of classes | Mapping year | Number of classes | Mapping year | |
| **Updated areas** | | | | | |
| CAF02 | 27 | 2015 | 21 | 2019 | 2998 |
| CAF07 | 17 | 2016 | 16 | 2019 | 3069 |
| CAF11 | 23 | 2016 | 19 | 2019 | 3228 |
| CAF99 | 17 | 2016 | 20 | 2019 | 2421 |
| **New areas** | | | | | |
| CAF05 | 24 | 2017 | 17 | 2019 | 4647 |
| | | | 17 | 2000 | 7168 |
| CAR01 | 29 | 2017 | 26 | 2000 | 4029 |
| EAF04 | 26 | 2017 | 18 | 2000 | 3943 |
| PAC01 | 28 | 2016 | 26 | 2000 | 4413 |
| | | | 30 | 2005 | |
| | | | 28 | 2010 | |



| SAF21 | 29 | 2017 | 18 | 2000 | 3995 |
| WAF04 | 24 | 2017 | 18 | 2000 | 3522 |

**Figure 3 Spatial distribution of the validation datasets within the updated key landscapes for conservation.**

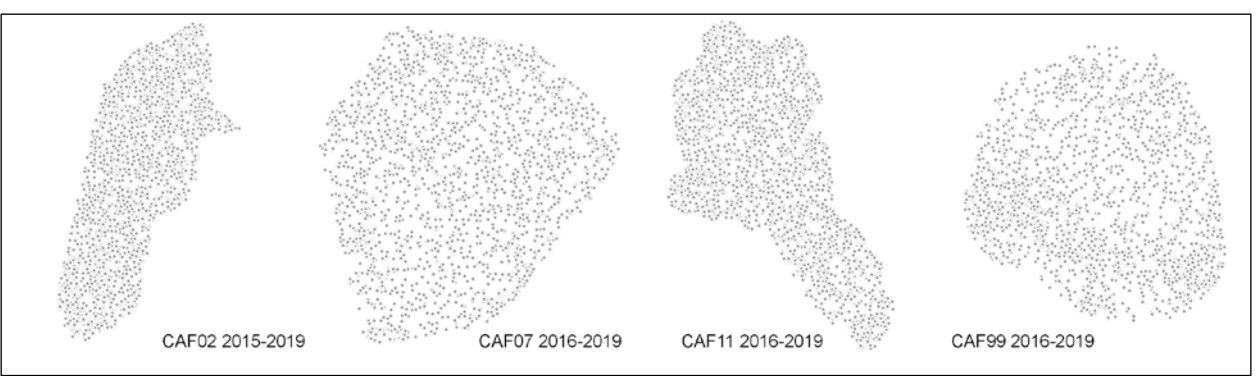

## 4    Data quality assessment

We updated some of the most critical landscapes (KLCs) due to various anthropogenic pressures for land cover change

compared to the base maps we presented in Szantoi and colleagues (2020). These KLCs were: Greater Virunga (CAF02), Salonga (CAF07), Upemba (CAF11), and Yangambi (CAF99). The Salonga KLC (CAF07) was mapped initially at the dichotomous LCCS level (Table 2, 8 land cover classes), but here we present both, the base map (2016) and a change map (2019), mapped at the modular LCCS level. The new land cover and land cover change maps (CAF05, CAR01, EAF04, PAC01, SAF21, and WAF04) were all mapped at the modular level for land cover as well as for change.

## 4.1 Technical Validation

*Spatial, temporal and logical consistency* was assessed by an independent procedure from the producer to determine the products positional accuracy, the validity of data with respect to time (seasonality), and the logical consistency of the data (topology, attribution and logical relationships). A Qualitative-systematic accuracy assessment was also performed wall-to-wall through a systematic visual examination for a) global thematic assessment b) expected size of polygons (Minimum

Mapping Unit (MMU)), c) seasonal effects and d) spatial patterns (i.e. following correct edges).

**Figure 4 Spatial distribution of the validation datasets within the new key landscapes for conservation.**



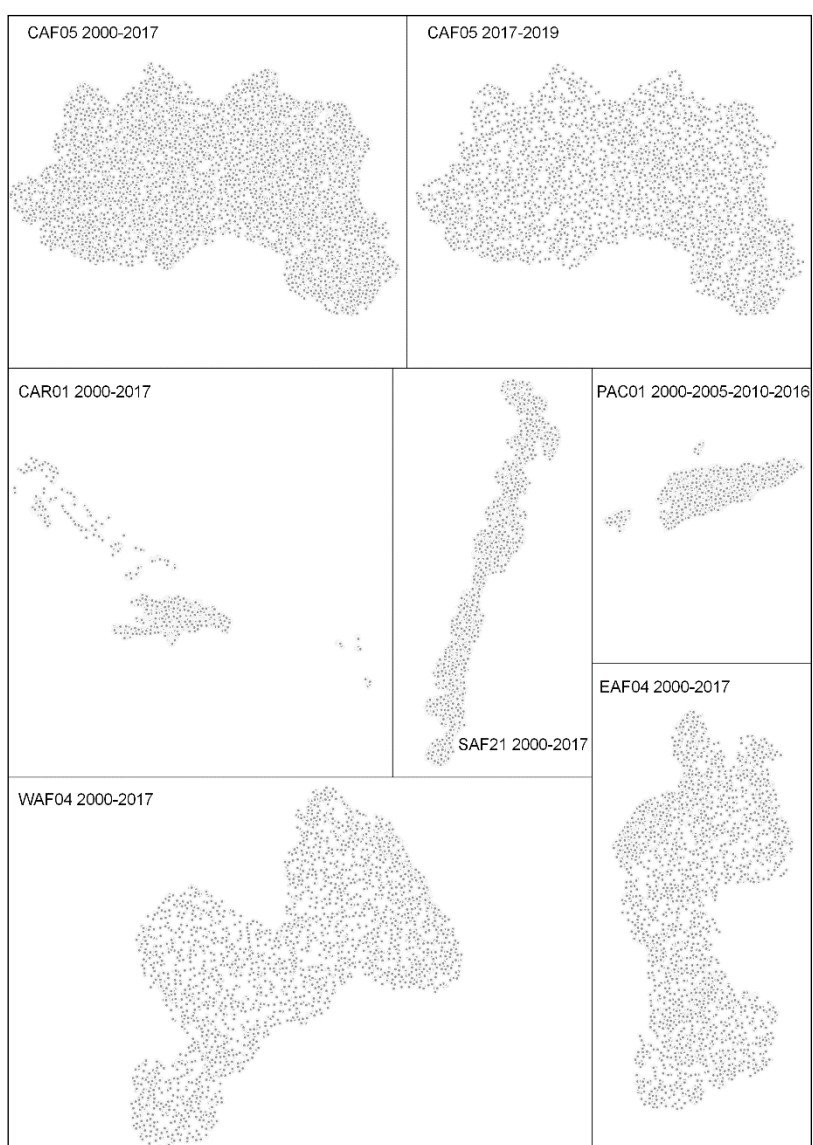

The quantitative accuracy assessment (i.e. validation) results are shown in Table 4 (overall accuracies), and in the Appendix (thematic class accuracies per KLC, Appendix A). Generally, the program aimed at a minimum of 85% overall accuracy for each product (KLC) and a minimum of 75% thematic accuracy (Producer's and User's) for each class within each KLC. The land cover change (LCC) accuracy should be >72%. In exceptional cases, the thematic accuracies might be lower than the threshold due to the difficulty to discriminate a particular class in a certain KLC.

Figure 5 shows the final LC and LCC products for the updated KLCs (CAF02, CAF07, CAF11, and CAF99) while Figures 6 (CAR01, WAF04), 7 (CAF05, EAF04, SAF21) and 8 (PAC01) show the new LC and LCC products, all classified at the



modular LCCS level. Some of the datasets presented in Figure 5 were already published in Earth System Science Data (Szantoi et al., 2020b): CAF02 year 2000 land cover change and year 2015 land cover maps; CAF07 year 2000 land cover change map; CAF11 year 2000 land cover change and year 2016 land cover maps; and CAF99 year 2000 land cover change and year 2016 land cover maps, for data access please see the Data Availability section.

**Table 4 Achieved overall accuracies for land cover mapping (%).**

| | LC map | Reference date | LCC map | Reference date |
|---|---|---|---|---|
| **Updated thematic maps** | | | | |
| CAF02 | 90.09 | 2015 | 99.38 | 2019 |
| CAF02 | 90.09 | 2015 | 91.93 | 2001 |
| CAF07 | 98.38 | 2016 | 98.36 | 2019 |
| CAF11 | 95.27 | 2016 | 95.81 | 2019 |
| CAF11 | 95.87 | 2016 | 95.81 | 2019 |
| CAF99 | 98.51 | 2016 | 99.31 | 2019 |
| CAF99 | 99.21 | 2016 | 99.31 | 2019 |
| **New thematic maps** | | | | |
| CAF05 | 90.63 | 2015 | 91.63 | 2019 |
| CAF05 | 91.75 | 2015 | 92.35 | 2000 |
| CAR01 | 92.55 | 2017 | 93.41 | 2000 |
| EAF04 | 97.30 | 2017 | 97.80 | 2000 |
| PAC01 | 91.28 | 2016 | 93.55 | 2000 |
| PAC01 | 91.28 | 2016 | 93.26 | 2005 |
| PAC01 | 91.28 | 2016 | 94.24 | 2010 |
| SAF21 | 91.00 | 2017 | 92.30 | 2000 |
| WAF04 | 97.20 | 2015 | 97.50 | 2000 |

LC - land cover, LCC - land cover change



**Figure 5 Key Landscapes for Conservation - modular classification level. The boundaries (black polygons) represent protected areas (IUCN category I-IV, UNEP-WCMC and IUCN, 2021) within the KLCs. Both land cover and land cover change maps are presented for each KLC.**

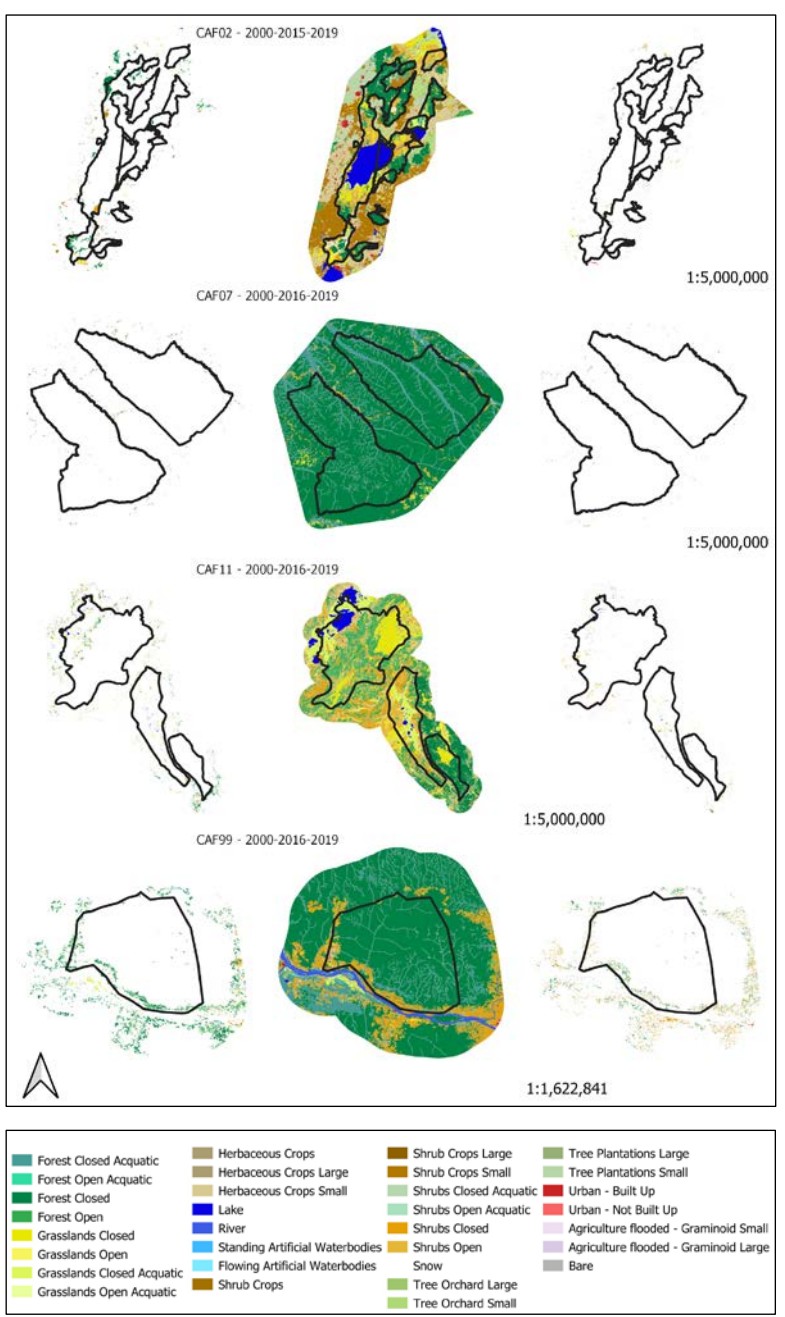

*CAF02 - Greater Virunga, CAF07 - Salonga, CAF11 - Upemba, CAF99 - Yangambi. Year 2000 datasets are available at (Szantoi et al., 2020b).



**Figure 6 Key landscapes for conservation - modular classification level. The boundaries (black polygons) represent protected areas**
**(IUCN category I-IV, UNEP-WCMC and IUCN, 2021) within the KLCs. Both land cover and land cover change maps are presented for each KLC. The inlets show the southeast part of the Caribbean KLC.**

*CAR01 - Caribbean, WAF04 - Wapok.

**Figure 7 Key Landscapes for Conservation - modular classification level. The boundaries (black polygons) represent protected areas (IUCN category I-IV, UNEP-WCMC and IUCN, 2021) within the KLCs. Both land cover and land cover change maps are presented for each KLC.**

Earth System
Science
Data

CAF05 - 2000-2017-2019

1:9,100,000

EAF04 - 2000-2017

SAF21 - 2000-2017

1:5,000,000

1:7,000,000

| | | | |
|---|---|---|---|
| ■ Forest Closed Acquatic | ■ Herbaceous Crops | ■ Shrub Crops Large | ■ Tree Plantations Large |
| ■ Forest Open Acquatic | ■ Herbaceous Crops Large | ■ Shrub Crops Small | ■ Tree Plantations Small |
| ■ Forest Closed | ■ Herbaceous Crops Small | ■ Shrubs Closed Acquatic | ■ Urban - Built Up |
| ■ Forest Open | ■ Lake | ■ Shrubs Open Acquatic | ■ Urban - Not Built Up |
| ■ Grasslands Closed | ■ River | ■ Shrubs Closed | ■ Agriculture flooded - Graminoid Small |
| ■ Grasslands Open | ■ Standing Artificial Waterbodies | ■ Shrubs Open | ■ Agriculture flooded - Graminoid Large |
| ■ Grasslands Closed Acquatic | ■ Flowing Artificial Waterbodies | Snow | ■ Bare |
| ■ Grasslands Open Acquatic | ■ Shrub Crops | ■ Tree Orchard Large | |
| | | ■ Tree Orchard Small | |

\* CAF05 - Garamba, EAF04 - Niassa Selous, SAF21 - Madagascar

**Figure 8 Timor-Leste Key Landscape for Conservation - modular classification level. The boundaries (black polygons) represent the country boundary. Both land cover and land cover change maps are presented for Timor-Leste.**

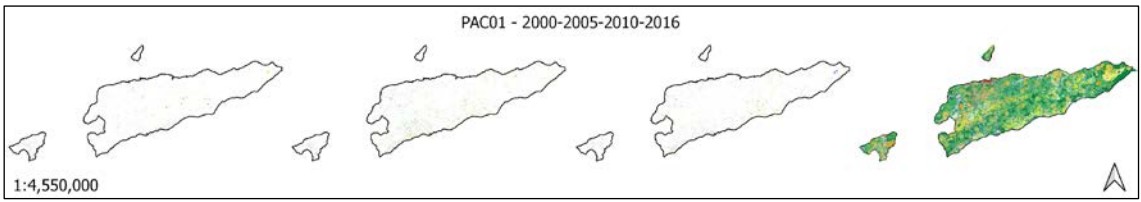

PAC01 - 2000-2005-2010-2016

1:4,550,000





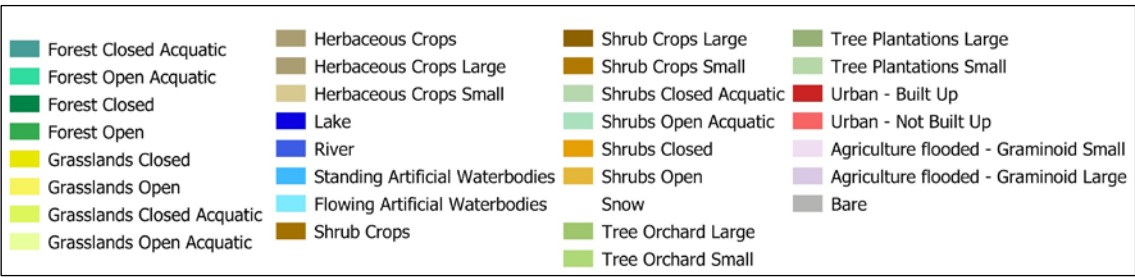

## 5    Discussion

There is a direct relationship between population growth, agricultural expansion, energy demand, and pressure on land. With the current state of development, population increase, and economic growth, a large portion of the sub-Saharan population

depends on the remaining natural resources to meet their food and energy needs (Brink et al., 2012), while in the Caribbean (CAR01) urbanization puts pressure on the natural resources (Nathaniel et al., 2021). In the case of Timor-Leste (PAC01) the peacebuilding process shapes the country's land cover and land use trends since 2006 (Ide et al., 2021). The demands of social and economic growth require additional land, typically at the expense of previously untouched areas. Areas under protection (i.e. national parks) that remain well-preserved (see Figs. 5, 6 and 7) often have regions in close proximity under tremendous

pressure. Such areas (many times transboundary ones) need very accurate monitoring and base maps, which are provided through this work, especially as areas shared between and/or among countries are frequently not mapped with a common legend, if mapped at all. The presented KLC datasets can be used for continuous land cover and land use monitoring, evaluation of management practices and effectiveness, endowment for scientific counsel, habitat modeling, information dissemination, and capacity building in their corresponding countries and to manage natural resources such as forests, soil, biodiversity,

ecosystem services, and agriculture (Tolessa et al., 2017). Furthermore, regional climate change, biogeochemical, and hydrologic models are currently capable of using high-resolution LC data for predictions in general (Nissan et al., 2019) and spatially focused (i.e. Africa) (Sylla et al., 2016; Vondou and Haensler, 2017).

The validation datasets are independently collected and verified through a robust procedure. Validation datasets can then be used for additional land cover mapping, creating spectral libraries, and the validation of other local, regional, and global

datasets. It is important that various land cover products can be used or compared against one another regardless of their geographic origins. Here, 10 land cover and land cover change maps for different areas in the OACPS where quality land cover products are missing (Marshall et al., 2017) were introduced. All data were produced using the unified Land Cover Classification System. The LCCS's modular level can be applied to local scales through its very detailed classes (here 32).



### 5.1 Drivers of change

Geist and Lambin (2002) describe the driving human forces of land cover changes as an interlinking of three key variables: expansion of agriculture, extraction of wood, and development of infrastructure (urbanization). The main land cover dynamic in sub-Saharan Africa can be explained by the first two variables, but increasingly with urbanization as well, just like in the other mapped areas (Caribbean, Timor-Leste) (Güneralp et al., 2017; Nathaniel et al., 2021; Hugo, 2019). Although the driving force behind the clearing of natural vegetation has traditionally been predominantly attributed to the expansion of new

agricultural land areas (including investments in large-scale commercial agriculture) (Brink and Eva, 2009), firewood extraction and charcoal production are also key factors in forest, woodland, and shrubland degradation throughout the region. This land cover dynamic is not just a by-product of greater forces such as logging for timber and agricultural expansion but stems from a specific need to satisfy energy demand (European Commission, 2018); in fact, in sub-Saharan Africa, the main use of extracted wood is for energy production (Kebede et al., 2010). Although the region possesses a huge diversity of energy

sources such as oil, gas, coal, uranium, and hydropower, the local infrastructure and use of these commercial energy sources are still somewhat limited. Traditional sources of energy in the form of firewood and charcoal account for over 75 % of the total energy use in the region (Kebede et al., 2010). Efforts to meet the population and economic demands in the OACPS while preserving biodiversity and ecosystem functioning require informed decision-making. The global component of the Copernicus Land Service (Copernicus Global Land), in particular the High-Resolution Hot Spot Monitoring component, presents a unique

opportunity for such information gathering.

### 5.2 Sources of errors

As the applied LCCS allows very detailed hierarchical classification, some classes can be difficult to distinguish from each other. This is especially true in Africa's vast and very heterogeneous landscapes where agricultural land use is mainly smallholder based (i.e., very small plots), while shifting cultivation is mostly due to the lack of fertilizers and weak soil, leading

to land abandonment. Landscapes are generally not composed of clearly fragmented and well-identifiable cover formation. In this region, landscapes usually form a continuum of various cover (vegetation) formations that might include different layers of tree, shrub, and herbaceous vegetation. These variations combined with differences in vegetation density (open vs. closed) and heights makes class assignments challenging. Moreover, some specific agriculture classes distinguish even the cultivation type, e.g., differentiating between fruit tree plantations and tree plantations for timber. Thus, the discrimination of such classes

is very difficult and might introduce classification errors. Apart from the land cover classification, errors could also be introduced due to climate-induced variability, such as leaf phenology where deciduous vegetation might appear bare during a dry period (season). At a more general level, difficulties in identifying between aquatic or regularly flooded surfaces and terrestrial areas have been observed in certain KLCs, especially when flooded periods are short.

As for Timor-Leste (PAC01), to discriminate between evergreen and deciduous natural vegetation was particularly challenging

across the seasonal variations.



Another specific source of error can be identified for the Caribbean KLC (CAR01), where the area consists of a vast complex of small islands (i.e. keys) and archipelagos that include large areas of coastal swamps. In these regions the connection of the coastal inland water surfaces with the open sea is often very difficult to be identified and consequently there are areas where the assignment of the water surface classes were ambiguous with respect to the open sea, that would result in the exclusion of

area from the map.

## 5.3 Current and future use of datasets

The C-HSM datasets have been widely used by policy makers (the Organisation of African, Caribbean and Pacific States (OACPS) and European partners) to help identify areas prone to change due to human activities. For example, COFED (Support Unit for the (DRC) National Authorizing Officer of the European Development Fund), the EEAS (European External

Action Service) of the DRC, manages an envelope of EUR 120 million, allocated for five protected areas in the DRC (Virunga, Garamba, Salonga, Upemba, and the Yangambi biosphere), where they use the C-HSM products for planning and for investment strategies (i.e., hydropower). Thus, the before mentioned PAs were requested to be updated in terms of land cover changes for 2019 by EEAS, which we present here in this study. Another example comes from West Africa, where nongovernmental organizations (NGOs, e.g., Wild Chimpanzee Foundation), public-benefit enterprises (i.e., German Society

for International Cooperation – GIZ), and national authorities (i.e., l'Office Ivoirien des Parcs et Réserves – OIPR) use the data to identify areas under pressure for agriculture (cocoa, oil palm, rubber, coconut) and human–wildlife conflicts in Cote d'Ivoire, Ghana, and Liberia. Additional areas (i.e. CAR01, PAC01) mapped and presented in this study can be used to help projects (e.g. BIOPAMA, https://biopama.org/) and countries to improve management and governance of their biodiversity and natural resources.

## 6    Data availability

The data are provided in a shapefile (*.shp) format, polygon geometry for the land cover and change datasets and point geometry for the validation datasets. The presented data are in the World Geodetic System 1984 geographic coordinate system (GCS) (EPSG:4326) and its datum (EPSG:6326). The validation data, besides using the same GCS, also have the Africa Albers equal-area conic (EPSG:102022) projection coordinate system.

Apart from CAF05 and PAC01, each KLCs is described by two polygon vector layers: a land cover (LC) layer and a land cover change (LCC) layer. In the case of CAF05, we present three layers (2000 and 2019 LCC and 2017 LC), and for PAC01 we present four layers (2000, 2005, and 2010 as LCC, and 2016 as LC). The LC layer is always a wall-to-wall map, covering the entire area of interest (AOI). The LC temporal reference for the project is the year 2016, although for each area the actual "mapping year" is noted in the file name (i.e., CAF05_2017) and generally refers to the year in which the largest number of

satellite images were used for the classification. The LCC layer provides a partial coverage of the AOI, as it contains only the



areas (polygons) where thematic change occurred compared to the LC layer. The LCC temporal reference is the year 2000 (± 3 years), noted in the file name (i.e., CAF05_2000).

Each LC and LCC shapefile comes with its corresponding attribute table, where two or three attributes are present: [map_codeA] – dichotomous class, [map_code] – modular class, [class_name] – corresponding modular class name.

Each of the 10 areas has been quantitatively validated using a spatially specific point dataset. These datasets were generated through the method described in section 3.5, and each point was used to verify the correctness of the LC–LCC maps. The corresponding data in the attribute table are LC – [plaus201X] and LCC – [plaus200X or plaus201X]. Both [plaus201X] and [plaus200X] attributes refer to the most detailed classification level attributes (map_code or map_codeA) present in the LC and LCC datasets (shapefiles). Some of the validation datasets contain only attributes of the aggregated classes, as described

in section 3.2, those attributes are named as [plaus201Xr, plaus200Xr].The plaus201X and plaus200X attributes refer to the year the validation sets represent, as these can be different among KLCs; the exact year is always noted in the columns' names (e.g., plaus2000, plaus2016).

The naming of all attributes follows the same structure in all data. Please see the details in the Appendix.

The complete package (all datasets together) is available for download at https://doi.org/10.5281/zenodo.4621375 (Szantoi et

al., 2021), or individually as source datasets (each KLC) from the same web address.

Besides archiving the datasets at Zenodo (www.zenodo.org) (last access: 22 March 2021) with corresponding digital object identifiers, the Copernicus High-Resolution Hot Spot Monitoring (C-HSM) website (https://land.copernicus.eu/global/hsm, last access: 22 March 2021) provides open access to all the land cover and land cover change presented in this article as well as technical reports and on-the-fly statistics.

**7    Conclusions**

The C-HSM service component is part of Copernicus Global Land, which produces near-real-time biophysical variables at medium scale, globally. In contrast, the C-HSM activity is an on-demand component that addresses specific user requests in the field of sustainable management of natural resources. The products presented here provide the second set of standardized land cover and land cover change datasets for 10 KLCs with their corresponding validation datasets in the African, Caribbean

and Pacific regions. The geographic distribution covers the tropical and subtropical regions of west, central, and southeastern Africa as well as a large part of the Caribbean region and Timor-Leste in the Pacific region. The most recent land cover change might be reassessed for selected already-mapped KLCs periodically in order to generate longer-term time series land cover dynamics information - as this is the case in the currently presented data (CAF02, CAF07, CAF11, and CAF99, see the original LC/LCC data in Szantoi et al., 2020). While this is not done systematically, but on specific customer requests, the C-HSM



service encourages stakeholder cooperation and provides capacity building workshops around the globe. In-person training events provide an opportunity for new and existing users to learn how to use and interpret data, operate the web information system, and easily assess recent land cover change data using Sentinel-2 image mosaics. Here, we provide very-high-quality products, which can be used directly as base maps and for policy decisions, as well as for comparison and/or evaluation of other land cover products or the implementation of validation datasets for training and validation purposes.

Finally, the service has a high degree of confidence that the data presented here (and in the previous phase, Szantoi et al., 2020) are of the highest quality, regularly reaching above 90 % overall accuracy. This is guaranteed by a rigorous and independent production and validation mechanism and feedback loop, which does not stop until the required overall and per-class accuracy levels are reached.

Following the general European Commission's Copernicus Programme open-access policy, the data are distributed free to any
user through a dedicated website (https://land.copernicus.eu/global/hsm, last access: 16 March 2021). This interactive online information system allows access to browse, analyze, and download the data, including the accuracy assessment information.

**Appendix**

Thematic class accuracies per KLC. Accuracy parameters are in percent, classes with less than 15 samples were not included in the overall accuracy calculation. Accuracy results are presented at the aggregated as well as at the modular LCCS levels,
depending on the type of mapping (land cover map - modular, or land cover change map - aggregated).

Class – corresponding class (see Table 2 "Modular" or "Aggregated" map code)

PA – producer's accuracy

UA – user's accuracy

NoRP – number of reference points

| CAF02 (aggregated) | | | | | | |
|---|---|---|---|---|---|---|
| | 2015 | | | 2019 | | |
| Class | PA | UA | NoRP | PA | UA | NoRP |
| 3 | 99.7 | 99.7 | 1277 | 99.7 | 99.6 | 1243 |
| 4 | 98.8 | 97.7 | 510 | 98.8 | 98.2 | 541 |
| 6 | 0 | 0 | 0 | 0 | 0 | 0 |
| 7 | 100 | 99 | 120 | 100 | 99 | 148 |
| 11 | 96.8 | 93.4 | 28 | 100 | 93.3 | 20 |
| 14 | 100 | 100 | 219 | 100 | 100 | 175 |
| 77 | 100 | 99.9 | 648 | 99.9 | 100 | 508 |
| 78 | 92.6 | 100 | 133 | 92.3 | 98.4 | 217 |





| 165 | 100 | 100 | 3 | 100 | 100 | 2 |
| 166 | 100 | 100 | 5 | 100 | 100 | 2 |
| 184 | 99.9 | 100 | 52 | 100 | 99.9 | 129 |
| 185 | 100 | 100 | 2 | 100 | 100 | 10 |

| CAF05 (aggregated) | | | | | | | | | |
|---|---|---|---|---|---|---|---|---|---|
| | 2000 | | | 2015 | | | 2019 | | |
| Class | PA | UA | NoRP | PA | UA | NoRP | PA | UA | NoRP |
| 3 | 92.8 | 76.9 | 396 | 85 | 92.4 | 249 | 85.9 | 89.6 | 211 |
| 4 | 91.4 | 95 | 2957 | 93.5 | 91.4 | 1720 | 93.4 | 91.3 | 1764 |
| 7 | 98.7 | 84.2 | 317 | 82.5 | 87.3 | 150 | 82.5 | 87.3 | 149 |
| 11 | 98.3 | 93.5 | 59 | 83.8 | 100 | 10 | 83.8 | 100 | 10 |
| 13 | 100 | 100 | 8 | 100 | 100 | 14 | 100 | 100 | 15 |
| 14 | 95.4 | 93.9 | 96 | 99.9 | 100 | 22 | 99.9 | 100 | 21 |
| 77 | 94.1 | 96.4 | 1956 | 94.8 | 96.2 | 1399 | 94.6 | 96.2 | 1283 |
| 78 | 90.7 | 83 | 1205 | 85.7 | 86.2 | 917 | 85.6 | 86.2 | 949 |
| 165 | 0 | 0 | 0 | 0 | 0 | 1 | 0 | 0 | 1 |
| 166 | 100 | 83.7 | 41 | 100 | 100 | 1 | 100 | 100 | 1 |
| 184 | 96.8 | 94.3 | 88 | 82.7 | 97.6 | 92 | 81.6 | 97.4 | 155 |
| 185 | 100 | 23.1 | 9 | 100 | 93.2 | 70 | 94.9 | 94 | 87 |

| CAF05 (all classes – LC map) | | | |
|---|---|---|---|
| 2015 | | | |
| Class | PA | UA | NoRP |
| 11 | 98.3 | 93.5 | 59 |
| 31 | 100 | 99.9 | 127 |
| 32 | 5.9 | 92.3 | 14 |
| 34 | 100 | 100 | 1 |
| 56 | 90 | 92.4 | 67 |
| 59 | 0 | 0 | 0 |
| 60 | 85.1 | 83 | 209 |
| 77 | 95.1 | 95.8 | 1954 |
| 78 | 89.9 | 82.8 | 1184 |
| 112 | 88.8 | 93.2 | 2355 |
| 116 | 81.2 | 74.9 | 285 |
| 148 | 72.6 | 84.2 | 215 |
| 152 | 94.4 | 93.6 | 9 |





| 165 | 0 | 0 | 0 |
|---|---|---|---|
| 166 | 100 | 85.1 | 40 |
| 171 | 98.4 | 73.7 | 82 |
| 175 | 98.8 | 95.6 | 75 |
| 178 | 98.1 | 87.2 | 152 |
| 182 | 87.5 | 28 | 8 |
| 184 | 95.1 | 95.8 | 161 |
| 185 | 100 | 100 | 50 |
| 187 | 100 | 100 | 8 |
| 190 | 95.4 | 94 | 80 |
| 191 | 100 | 95.8 | 23 |

| CAF07 (all classes – LC/LCC map) | | | | | | | |
|---|---|---|---|---|---|---|---|
| 2016 | | | | 2019 | | | |
| Class | PA | UA | NoRP | Class | PA | UA | NoRP |
| 11 | 100 | 100 | 2 | 11 | 100 | 100 | 2 |
| 31 | 96.6 | 83.6 | 53 | 31 | 95.9 | 84.2 | 52 |
| 32 | 96.4 | 66.7 | 3 | 32 | 97.6 | 33.3 | 4 |
| 56 | 95.1 | 77.5 | 91 | 56 | 87.8 | 75.8 | 112 |
| 60 | 91.3 | 89.8 | 102 | 60 | 91.3 | 72.6 | 89 |
| 77 | 98.4 | 99.8 | 1605 | 77 | 98.5 | 99.8 | 1524 |
| 78 | 82.7 | 92.7 | 98 | 78 | 90.1 | 94.9 | 124 |
| 112 | 89.5 | 86.1 | 231 | 112 | 89 | 88.6 | 297 |
| 116 | 96.2 | 96.8 | 61 | 116 | 82.8 | 90 | 30 |
| 148 | 99.8 | 97.4 | 134 | 148 | 99.4 | 97.5 | 144 |
| 165 | 99.3 | 92.3 | 386 | 152 | 0 | 0 | 0 |
| 166 | 31.6 | 75 | 19 | 165 | 99.3 | 92.3 | 379 |
| 171 | 94.1 | 94.3 | 54 | 166 | 31.6 | 47.2 | 19 |
| 175 | 0 | 0 | 2 | 171 | 94.5 | 94 | 65 |
| 178 | 100 | 85 | 51 | 175 | 50 | 100 | 4 |
| 184 | 83.1 | 90.4 | 77 | 178 | 92.1 | 85.4 | 38 |
| 190 | 87.8 | 93.8 | 77 | 184 | 81 | 90.5 | 87 |
| 191 | 100 | 100 | 22 | 190 | 87.7 | 92.6 | 76 |
| | | | | 191 | 100 | 100 | 22 |

| CAF11 (aggregated) |
|---|

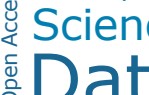

| | 2000 | | | 2016 | | | 2019 | | |
|---|---|---|---|---|---|---|---|---|---|
| Class | PA | UA | NoRP | PA | UA | NoRP | PA | UA | NoRP |
| 3 | 98.7 | 92.8 | 339 | 92.9 | 95.1 | 201 | 93 | 96.2 | 272 |
| 4 | 99.3 | 93.8 | 1169 | 99.2 | 92.4 | 1099 | 99.2 | 92.2 | 999 |
| 6 | 100 | 14.4 | 2 | 42.4 | 100 | 33 | 42.5 | 100 | 33 |
| 7 | 96.9 | 99.2 | 614 | 97.8 | 96.5 | 373 | 97.9 | 96.8 | 372 |
| 11 | 100 | 96.7 | 30 | 0 | 0 | 0 | 0 | 0 | 0 |
| 14 | 98.7 | 99.9 | 275 | 99.8 | 99.4 | 120 | 100 | 99.8 | 111 |
| 77 | 94.5 | 95.6 | 529 | 90.5 | 98.9 | 515 | 90.4 | 98.8 | 430 |
| 78 | 92.6 | 97.7 | 597 | 95 | 98.4 | 711 | 94.8 | 98.3 | 760 |
| 165 | 79.4 | 96.3 | 79 | 77.1 | 100 | 7 | 77 | 100 | 5 |
| 166 | 98.7 | 99.2 | 47 | 99.8 | 99.3 | 12 | 99.8 | 99.2 | 11 |
| 184 | 100 | 95.8 | 87 | 99.9 | 94.6 | 81 | 100 | 94.9 | 157 |
| 185 | 100 | 95.4 | 17 | 100 | 100 | 76 | 93.8 | 100 | 78 |

| CAF11 (all classes – LC map) | | | |
|---|---|---|---|
| 2015 | | | |
| Class | PA | UA | NoRP |
| 11 | 100 | 100 | 30 |
| 32 | 100 | 100 | 26 |
| 34 | 0 | 0 | 0 |
| 56 | 69.9 | 100 | 1 |
| 59 | 92.4 | 99.1 | 74 |
| 60 | 97.3 | 97.1 | 339 |
| 77 | 94.6 | 95.2 | 488 |
| 78 | 92.4 | 97.1 | 534 |
| 112 | 96.8 | 86.9 | 441 |
| 116 | 97.7 | 94.3 | 289 |
| 148 | 98.5 | 97.1 | 325 |
| 152 | 0 | 0 | 0 |
| 160 | 100 | 100 | 3 |
| 165 | 79.1 | 96.2 | 78 |
| 166 | 96.9 | 99.2 | 46 |
| 171 | 75 | 92.7 | 74 |
| 175 | 56.8 | 98.6 | 72 |
| 178 | 97.9 | 98 | 411 |
| 182 | 95 | 95 | 20 |
| 184 | 100 | 98.9 | 167 |
| 185 | 100 | 100 | 75 |





| 190 | 87.9 | 98.2 | 90 |
| 191 | 99.8 | 100 | 202 |

| CAF99 (aggregated) | | | | | | | | | |
|---|---|---|---|---|---|---|---|---|---|
| | 2000 | | | 2016 | | | 2019 | | |
| Class | PA | UA | NoRP | PA | UA | NoRP | PA | UA | NoRP |
| 3 | 91.6 | 98.9 | 431 | 85.9 | 98 | 241 | 86.2 | 98.7 | 193 |
| 4 | 92.4 | 92.1 | 417 | 98.4 | 96.4 | 397 | 99.5 | 97.5 | 452 |
| 7 | 100 | 97.8 | 231 | 99.8 | 88 | 72 | 94.7 | 88.8 | 76 |
| 14 | 100 | 100 | 175 | 100 | 100 | 108 | 100 | 100 | 109 |
| 77 | 99 | 99.2 | 905 | 99.7 | 99.9 | 1139 | 99.7 | 99.9 | 1098 |
| 78 | 93.6 | 85.1 | 210 | 97 | 99.8 | 60 | 92.1 | 93.1 | 43 |
| 165 | 97.8 | 97.9 | 246 | 100 | 99.1 | 352 | 100 | 99.1 | 346 |
| 166 | 100 | 88.7 | 40 | 100 | 82.2 | 22 | 99.8 | 81.6 | 16 |
| 184 | 99.4 | 88.3 | 72 | 99.4 | 100 | 28 | 98.7 | 99.8 | 85 |
| 185 | 0 | 0 | 0 | 0 | 0 | 0 | 0 | 0 | 0 |

| CAF99 (all classes – LC map) | | | |
|---|---|---|---|
| 2015 | | | |
| Class | PA | UA | NoRP |
| 31 | 91.6 | 99.8 | 267 |
| 32 | 94.5 | 100 | 69 |
| 56 | 100 | 99.5 | 76 |
| 59 | 100 | 9.5 | 4 |
| 60 | 91.9 | 96.5 | 125 |
| 77 | 99.6 | 99.2 | 732 |
| 78 | 79.1 | 91.5 | 156 |
| 112 | 96.1 | 95.9 | 341 |
| 148 | 98.7 | 96.9 | 168 |
| 165 | 97.8 | 97.5 | 240 |
| 166 | 100 | 89.2 | 42 |
| 171 | 100 | 100 | 102 |
| 175 | 0 | 0 | 3 |
| 178 | 100 | 91.6 | 77 |
| 184 | 100 | 95.9 | 150 |
| 185 | 100 | 100 | 2 |
| 190 | 100 | 100 | 113 |



 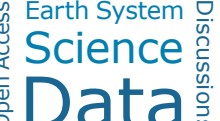

| 191 | 100 | 100 | 60 |
|---|---|---|---|

| CAR01 | | | | | | | |
|---|---|---|---|---|---|---|---|
| Aggregated classes | | | | All classes – LC map | | | |
| 2000 | | | | 2017 | | | |
| Class | PA | UA | NoRP | Class | PA | UA | NoRP |
| 3 | 90.8 | 94.5 | 874 | 11 | 91.9 | 86.5 | 79 |
| 4 | 90.1 | 96.1 | 890 | 31 | 83.1 | 83.2 | 110 |
| 6 | 98.8 | 97.3 | 160 | 32 | 98.9 | 84.5 | 65 |
| 7 | 93 | 92.1 | 343 | 33 | 80.6 | 79.8 | 65 |
| 11 | 83.7 | 82.7 | 70 | 34 | 100 | 81.9 | 24 |
| 13 | 99.8 | 83.5 | 155 | 55 | 98.3 | 86.2 | 71 |
| 14 | 89.7 | 93.6 | 181 | 56 | 100 | 92.9 | 87 |
| 77 | 97.9 | 90.6 | 519 | 59 | 91 | 92.3 | 159 |
| 78 | 92.5 | 88.6 | 346 | 60 | 85.8 | 92.2 | 272 |
| 165 | 96 | 89.7 | 61 | 77 | 97.8 | 93.3 | 513 |
| 166 | 100 | 92.3 | 57 | 78 | 89.4 | 88.5 | 332 |
| 184 | 92.5 | 98.1 | 122 | 112 | 90.4 | 93.4 | 379 |
| 185 | 100 | 97.2 | 64 | 116 | 92.3 | 94.6 | 116 |
| 999 | 99.6 | 98.2 | 173 | 148 | 88.5 | 89.5 | 270 |
| | | | | 152 | 100 | 92.8 | 63 |
| | | | | 159 | 96 | 97.5 | 81 |
| | | | | 160 | 82.1 | 97.5 | 85 |
| | | | | 165 | 94.8 | 89.6 | 63 |
| | | | | 166 | 100 | 91.8 | 56 |
| | | | | 171 | 90.7 | 90.9 | 102 |
| | | | | 175 | 93.4 | 95.3 | 85 |
| | | | | 178 | 95.5 | 84.6 | 92 |
| | | | | 182 | 98.9 | 82.6 | 58 |
| | | | | 184 | 92.2 | 99.8 | 209 |
| | | | | 185 | 100 | 97 | 75 |
| | | | | 186 | 96.2 | 93.3 | 71 |
| | | | | 187 | 97.6 | 87.5 | 81 |
| | | | | 190 | 97.5 | 92.7 | 79 |
| | | | | 191 | 87 | 100 | 112 |
| | | | | 999 | 99.7 | 98.2 | 172 |





| EAF04 | | | | | | | |
|---|---|---|---|---|---|---|---|
| Aggregated classes | | | | All classes – LC map | | | |
| 2000 | | | | 2017 | | | |
| Class | PA | UA | NoRP | Class | PA | UA | NoRP |
| 3 | 93.4 | 95 | 638 | 11 | 100 | 98.7 | 86 |
| 4 | 96.8 | 96.3 | 834 | 31 | 100 | 79.4 | 43 |
| 6 | 83 | 82.1 | 130 | 32 | 100 | 100 | 12 |
| 7 | 92.4 | 95.7 | 260 | 33 | 100 | 97.6 | 129 |
| 11 | 100 | 98.7 | 86 | 34 | 90.9 | 99.6 | 97 |
| 14 | 99.5 | 97.9 | 172 | 55 | 100 | 99.8 | 78 |
| 77 | 99.3 | 98.5 | 952 | 56 | 100 | 93.8 | 30 |
| 78 | 97.3 | 98.5 | 723 | 59 | 100 | 100 | 82 |
| 165 | 100 | 100 | 51 | 60 | 96.8 | 94.4 | 269 |
| 166 | 0 | 0 | 2 | 77 | 98.8 | 98 | 922 |
| 184 | 99.6 | 97.4 | 90 | 78 | 96.6 | 98.4 | 652 |
| 185 | 100 | 83.3 | 5 | 112 | 95.6 | 95.1 | 465 |
| | | | | 116 | 91.3 | 97.8 | 114 |
| | | | | 148 | 99.7 | 94.8 | 135 |
| | | | | 152 | 100 | 77.3 | 17 |
| | | | | 159 | 0 | 0 | 0 |
| | | | | 160 | 93.7 | 99.5 | 138 |
| | | | | 165 | 100 | 100 | 51 |
| | | | | 166 | 0 | 0 | 2 |
| | | | | 171 | 100 | 91 | 35 |
| | | | | 175 | 60.9 | 83.4 | 11 |
| | | | | 178 | 92.3 | 95.1 | 211 |
| | | | | 184 | 99.8 | 100 | 171 |
| | | | | 185 | 100 | 92 | 23 |
| | | | | 190 | 99.8 | 98.9 | 92 |
| | | | | 191 | 100 | 98.5 | 78 |

| PAC01 (aggregated classes) | | | | | | | | | | | |
|---|---|---|---|---|---|---|---|---|---|---|---|
| 2000 | | | | 2005 | | | | 2010 | | | |
| Class | PA | UA | NoRP | Class | PA | UA | NoRP | Class | PA | UA | NoRP |

| | | | | | | | | | | | |
|---|---|---|---|---|---|---|---|---|---|---|---|
| 3 | 89.6 | 89.5 | 603 | 3 | 87.9 | 89.4 | 602 | 3 | 92.2 | 91.5 | 600 |
| 4 | 88.2 | 96.3 | 983 | 4 | 88 | 96.2 | 967 | 4 | 92 | 95.4 | 908 |
| 6 | 95.9 | 93.9 | 158 | 6 | 95.7 | 94.7 | 147 | 6 | 94 | 93.6 | 151 |
| 7 | 96.2 | 96.4 | 380 | 7 | 95.6 | 96 | 361 | 7 | 93.6 | 93.9 | 341 |
| 11 | 81.1 | 88.2 | 86 | 11 | 97.7 | 88 | 81 | 11 | 93.5 | 88.2 | 87 |
| 13 | 94.1 | 88.9 | 34 | 13 | 94.2 | 86.7 | 35 | 13 | 96.4 | 93 | 38 |
| 14 | 90.4 | 93.9 | 269 | 14 | 91 | 94.8 | 303 | 14 | 91.1 | 94.8 | 334 |
| 77 | 98.2 | 91.8 | 713 | 77 | 98.2 | 91.2 | 707 | 77 | 97.5 | 93.5 | 722 |
| 78 | 92.4 | 95 | 821 | 78 | 91.8 | 94.7 | 805 | 78 | 92.3 | 95.3 | 811 |
| 165 | 92.6 | 93.7 | 88 | 165 | 89.8 | 94.2 | 87 | 165 | 92.9 | 93 | 75 |
| 166 | 93.2 | 99.2 | 78 | 166 | 90.8 | 98.8 | 75 | 166 | 96.7 | 98.8 | 72 |
| 184 | 94.3 | 91.7 | 120 | 184 | 94.4 | 93 | 163 | 184 | 95 | 96 | 190 |
| 185 | 100 | 94.9 | 12 | 185 | 100 | 95.1 | 13 | 185 | 97.3 | 100 | 17 |
| 999 | 96.3 | 78 | 61 | 999 | 96.3 | 78 | 61 | 999 | 96.3 | 78 | 61 |

| PAC01 (all classes – LC map) | | | |
|---|---|---|---|
| 2016 | | | |
| Class | PA | UA | NoRP |
| 11 | 96.4 | 91.1 | 89 |
| 31 | 87.2 | 96.8 | 70 |
| 32 | 94.5 | 85.2 | 50 |
| 33 | 0 | 0 | 1 |
| 34 | 0 | 0 | 1 |
| 55 | 60.8 | 100 | 13 |
| 56 | 99.2 | 96.4 | 29 |
| 60 | 93.1 | 88.1 | 386 |
| 91 | 95.8 | 90.8 | 536 |
| 92 | 83.2 | 87.5 | 236 |
| 95 | 96.5 | 89.2 | 390 |
| 96 | 84.6 | 95.9 | 423 |
| 123 | 89.3 | 78.8 | 132 |
| 124 | 88.9 | 97.8 | 160 |
| 139 | 98.9 | 87.2 | 100 |
| 140 | 96.3 | 89.9 | 113 |
| 148 | 89.5 | 94 | 356 |
| 152 | 0 | 0 | 3 |
| 160 | 92.1 | 94.4 | 140 |
| 165 | 94.1 | 90.4 | 78 |
| 166 | 89 | 98.7 | 75 |





| | | | |
|---|---|---|---|
| 171 | 98.4 | 93.4 | 53 |
| 175 | 98.3 | 92.9 | 72 |
| 178 | 95.5 | 95.3 | 212 |
| 182 | 100 | 95.7 | 14 |
| 184 | 91.7 | 96.1 | 234 |
| 185 | 96.3 | 100 | 23 |
| 187 | 96 | 95.3 | 44 |
| 190 | 88.7 | 94.3 | 277 |
| 191 | 100 | 97.3 | 29 |
| 999 | 96.3 | 78 | 61 |

| SAF21 | | | | | | | |
|---|---|---|---|---|---|---|---|
| Aggregated classes | | | | All classes – LC map | | | |
| 2000 | | | | 2017 | | | |
| Class | PA | UA | NoRP | Class | PA | UA | NoRP |
| 3 | 89.5 | 84 | 517 | 11 | 95.3 | 92.8 | 67 |
| 4 | 94.9 | 92.4 | 1352 | 31 | 83.8 | 91.6 | 110 |
| 6 | 75.2 | 80.6 | 269 | 32 | 2.5 | 30.4 | 14 |
| 7 | 84 | 82.7 | 238 | 33 | 25 | 100 | 12 |
| 11 | 95.3 | 94.2 | 68 | 34 | 99.7 | 96.5 | 69 |
| 13 | 89.2 | 98 | 140 | 55 | 98.8 | 97.3 | 75 |
| 14 | 83.2 | 96.4 | 176 | 56 | 100 | 34.1 | 14 |
| 77 | 93 | 97.2 | 856 | 59 | 98.3 | 98.2 | 59 |
| 78 | 87.8 | 82.2 | 228 | 60 | 88.3 | 82.6 | 179 |
| 165 | 100 | 11.9 | 5 | 77 | 94.4 | 96.4 | 692 |
| 166 | 0.4 | 16.7 | 13 | 78 | 88 | 81.8 | 253 |
| 184 | 100 | 76.4 | 81 | 112 | 93 | 88.4 | 725 |
| 185 | 96 | 94.1 | 50 | 116 | 94.3 | 80.7 | 79 |
| 999 | 0 | 0 | 1 | 148 | 89.8 | 93.8 | 530 |
| | | | | 152 | 84.7 | 85.4 | 47 |
| | | | | 156 | 0 | 0 | 1 |
| | | | | 159 | 100 | 14.7 | 5 |
| | | | | 160 | 76 | 81.5 | 273 |
| | | | | 165 | 100 | 11.9 | 5 |
| | | | | 166 | 0.4 | 16.7 | 13 |
| | | | | 171 | 100 | 79.1 | 84 |
| | | | | 175 | 67.6 | 96.6 | 19 |





| | | | | 178 | 85.5 | 83.5 | 125 |
|---|---|---|---|---|---|---|---|
| | | | | 182 | 12.9 | 66.7 | 3 |
| | | | | 184 | 100 | 94.5 | 153 |
| | | | | 185 | 99.7 | 99.4 | 72 |
| | | | | 186 | 100 | 94.1 | 64 |
| | | | | 187 | 87.9 | 98.6 | 76 |
| | | | | 190 | 79.7 | 97.6 | 99 |
| | | | | 191 | 95.4 | 93.3 | 76 |
| | | | | 999 | 0 | 0 | 1 |

| WAF04 | | | | | | | |
|---|---|---|---|---|---|---|---|
| Aggregated classes | | | | All classes – LC map | | | |
| 2000 | | | | 2015 | | | |
| Class | PA | UA | NoRP | Class | PA | UA | NoRP |
| 3 | 99.5 | 93.7 | 670 | 11 | 100 | 100 | 48 |
| 4 | 97.4 | 98.8 | 1345 | 31 | 100 | 100 | 9 |
| 6 | 91.7 | 84.5 | 67 | 32 | 80 | 100 | 5 |
| 7 | 98.6 | 95.3 | 239 | 33 | 92.8 | 100 | 17 |
| 11 | 100 | 100 | 47 | 34 | 99.1 | 99 | 75 |
| 13 | 97 | 100 | 108 | 60 | 99.5 | 98.1 | 726 |
| 14 | 97.7 | 97.3 | 162 | 77 | 97.9 | 95.2 | 146 |
| 77 | 95.5 | 97.4 | 151 | 78 | 97.1 | 98.3 | 487 |
| 78 | 96 | 98.2 | 537 | 112 | 98.3 | 96.3 | 756 |
| 165 | 100 | 73.3 | 21 | 116 | 86.1 | 98.1 | 297 |
| 166 | 98.6 | 93.7 | 60 | 148 | 83.6 | 98.9 | 90 |
| 184 | 100 | 97.5 | 83 | 152 | 98.7 | 99.5 | 40 |
| 185 | 100 | 100 | 8 | 160 | 81.8 | 89 | 82 |
| | | | | 165 | 100 | 72.4 | 20 |
| | | | | 166 | 98.5 | 92.5 | 59 |
| | | | | 171 | 92.7 | 95 | 59 |
| | | | | 175 | 96.5 | 98.6 | 32 |
| | | | | 178 | 97.3 | 72.5 | 142 |
| | | | | 182 | 100 | 97.5 | 29 |
| | | | | 184 | 100 | 97.8 | 151 |
| | | | | 185 | 100 | 100 | 10 |
| | | | | 187 | 100 | 100 | 79 |
| | | | | 190 | 97.6 | 98.7 | 79 |





| | | | | 191 | 97.7 | 97.3 | 70 |
|---|---|---|---|---|---|---|---|

## Author contributions

ZSZ, ABB, and AL designed the work and wrote the paper.

## Competing interests

The authors declare that they have no conflict of interest.

## Disclaimer

All features and data are provided "as is" with no warranties of any kind.

## Acknowledgements

The development of the thematic maps has been made possible thanks to the effort of eGEOS – an Italian Space Agency (ASI)/Telespazio Company, ITHACA (Information Technology for Humanitarian Assistance, Cooperation and Action) and Telespazio – a Leonardo and Thales company; their quality evaluations were made possible by IGNFI (France), Joanneum Research (Austria), EOXPLORE (Germany), GISBOX (Romania), Space4environment (Luxembourg), ONFI (France), and LuxSpace (Luxembourg). This work was produced under the European Commission Copernicus program, Global Land Service, High Resolution Hot-Spot Monitoring component.

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
