# Peer review of "An update and beyond: key landscapes for conservation land cover and change monitoring, thematic and validation datasets for the African, Caribbean and Pacific regions"

_Earth System Science Data, 2021_

## Referee Comment (RC1)

The follow-on study by Szantoi et al. titled "An update and beyond: key landscapes for conservation land cover and change monitoring, thematic and validation datasets for the African, Caribbean and Pacific region" is a valuable and welcomed study. The manuscript is well written, albeit several grammar and typographical/formatting transgressions. The following questions/comments is noted:

L53-55: why use SPOT data for only TL? Why not use SPOT data for all areas and/or S-2 and Landsat for all areas? Please explain/clarify.

Study area: what is the reasoning for selecting the four sites (phase 1) for remapping? Why not remap all phase 1 sites?

Table 1: formatting of values in "Area" column

L84: please clarify the statement "optional topographic correction".

L86-88: incoherent and not easily interpretable. Please rewrite/restructure sentences for clarity.

L100: "Automatic". Please clarify how the classification process is automatic/automated.

L101: please clarify the statement "pre-selected imagery". Were not all the data used?

L102: please clarify the need to "reduce data dimensionality" considering that all the data together with the VI were processed as a data cube.

L110: please clarify the statement "All the pre-processed data". Seems redundant considering that all data were pre-processed and used in the analysis?

L120-134: text is vague and incoherent. Please rewrite/restructure text for cohesion and coherence.

L194-196: what is the basis for deciding on the minimum overall, thematic, and LCC accuracy?

"Discussion": avoid superfluous language, e.g. adjectives such as "tremendous".

L248: provide argument for the robustness of the validation process as described in L156-161. How is the process robust and perhaps more importantly repeatable, considering the inherent human error/bias? What are the alternatives to ensuring robustness and repeatability?

"Sources of errors": what of bias and uncertainty?

L308-309: why not re-project all data to a common projected coordinate system?

L320-321: how robust and repeatable is this process?

---

## Author Comment (AC1)

**RESPONSE TO REFEREE #1**

**The follow-on study by Szantoi et al. titled "An update and beyond: key landscapes for conservation land cover and change monitoring, thematic and validation datasets for the African, Caribbean and Pacific region" is a valuable and welcomed study.**

Dear Referee #1 – we thank you for the positive review.

**The manuscript is well written, albeit several grammar and typographical/formatting transgressions.**

The entire manuscript was edited for grammar and typographical/formatting transgressions.

The following questions/comments are noted:

**L53-55: why use SPOT data for only TL? Why not use SPOT data for all areas and/or S-2 and Landsat for all areas? Please explain/clarify.**

We rewrote the text to improve clarity:

"The datasets are based mainly on freely available medium spatial resolution data: Copernicus Sentinel-2 (S-2) data for maps after 2015, and United States Geological Survey Landsat 7 and 8 (LS7, LS8) data for maps before 2015. The exceptions are three areas (Caribbean, Timor Leste and Wapok) where we used *Centre national d'études spatiales* SPOT (SP4, SP5, SP6) data, because S-2 and LS7/8 had limited coverage for the time period we mapped."

**Study area: what is the reasoning for selecting the four sites (phase 1) for remapping? Why not remap all phase 1 sites?**

We added a paragraph to clarify:

"We selected four previously mapped KLCs (Szantoi et al., 2020b) to be remapped: Salonga (CAF07) because of the less detailed initial mapping (LCCS dichotomous level only), and Greater Virunga (CAF02), Upemba (CAF11) and Yangambi because of site importance identified by the BIOPAMA Programme and the Delegation of the European Union to DR Congo."

**Table 1: formatting of values in "Area" column**

Done

**L84: please clarify the statement "optional topographic correction".**

The statement was revised:

"The Shuttle Radar Topography Mission (SRTM, 30 m or 90 m) digital elevation model was used to estimate the target height and slope, as well as the surface sun incidence angles to apply topographic correction."

**L86-88: incoherent and not easily interpretable. Please rewrite/restructure sentences for clarity.**

The sentences were revised as follows:

"Additionally, as satellite data were limited for some of the mapped areas, especially for the years 2000 and 2005, imagery was collected for a target year (e.g. 2000) ± 3 years. In some cases, this was expanded to ± 5 years, or to where four cloud-free observations per pixel had been collected for the specified date and location."

**L100: "Automatic". Please clarify how the classification process is automatic/automated.**

We replaced the phrase "Automatic classification" with "Image classification" as it better describes the procedure. The "automatic" originally was referring to the Support Vector Machine classifier, but in fact, the entire process is not fully automatic (e.g. various indices generation).

**L101: please clarify the statement "pre-selected imagery". Were not all the data used? and**
**L102: please clarify the need to "reduce data dimensionality" considering that all the data together with the VI were processed as a data cube.**

The statement was updated, as we used all the data.

"Based on the imagery data (Appendix A), dense multitemporal timeseries (DMT) were generated to allow proper characterisation of the temporal variability of the spectral features through various vegetation indices, aiding the LC class labelling process."

**L110: please clarify the statement "All the pre-processed data". Seems redundant considering that all data were pre-processed and used in the analysis?**

Indeed, all the data were "pre-proceesd", thus, we updated the sentence as follows:

"Imagery data (spectral bands and vegetation indices) were fed into the Support Vector Machine (SVM) supervised classification model."

**L120-134: text is vague and incoherent. Please rewrite/restructure text for cohesion and coherence.**

We rewrote the text to improve clarity:

[revised manuscript text omitted]

**L308-309: why not re-project all data to a common projected coordinate system?**

We think that there are different user needs – ie. for global analysis WGS84 will work, for local analysis the user will re-project the data to the most appropriate projected coordinate system.

**L320-321: how robust and repeatable is this process?**

Here, we would refer back to the previous comment (L248) and the added paragraph:

"The validation datasets are independently collected and verified through a robust procedure. The entire product validation procedure is systematically repeatable, as it includes three separate components that are independently assessed: (1) the spatial, temporal and logical consistency component, (2) the qualitative accuracy component, and (3) the quantitative accuracy component. Each of these components

can be performed separately, with the use of standardised informatics tools. In particular, the quantitative assessment validation component is structured with a sequence of steps in which interpretation of the LC classes is iterated when a cover (or change) is in doubt. Furthermore, a random quality check of the interpretation is performed on 10% of the interpretation points."

**RESPONSE TO REFEREE #2**

**The article is very relevant in that it is an update to a previous data production effort by the same author and his colleagues aimed at addressing the lack of up-to-date data for natural resource managers in developing countries.**

Dear Referee #2, we truly appreciate your time and effort.

**The methodology as described looks very sound and easy to understand but I was wondering why the authors did not use the Landsat surface reflectance product but rather the Level 1 product. An explanation in the text would be useful.**

We added/updated the corresponding paragraph as follows:

"Landsat ETM+ and OLI at Level1TP, Sentinel-2 at Level1C, and SPOT 4, 5 and 6 at Level1-B imagery were used in producing and updating the land cover and change maps. As we previously developed a surface reflectance production chain in our workflow (Szantoi et al., 2020b), the Level1TP (Landsat), Level1C (Sentinel-2), and Level1-B (SPOT) data were further corrected for atmospheric conditions to produce such products for the classification phase. The atmospheric correction module was implemented based on the 6S direct radiative transfer model for Landsat (Masek et al., 2006) and SPOT (Haifeng et al., 2010), and using the Sen2Cor processor (v2.8) based on the ATCOR model (Richter et al., 2012)."

**It would also be useful to specify the image dates of images used for the mapping in Table 1. This will help inform others who would want to replicate the methodology know which acquisition windows to explore for image data selection. Similarly, the independent map validation and accuracy assessment look sound, making the datasets very useful for effective site-specific assessments and monitoring**

Excellent point. While the data would not fit in Table 1, we decided to create a new table in the Appendix (Appendix A):

**A. Satellite data collecting period and type used for LC and LCC mapping**

| KLC | LC map | Data period | Data type* | LCC map | Data period | Data type* |
|---|---|---|---|---|---|---|
| CAF02 | 2015 | 07/2013 - 10/2016 | LS8 | 2019 | 01/2019 - 12/2019 | S-2 |
| CAF07 | 2016 | 05/2013 - 10/2016 | LS8 | 2019 | 01/2019 - 01/2020 | S-2 |
| CAF11 | 2016 | 01/2015 - 06/2016 | LS8 | 2019 | 01/2019 - 10/2019 | S-2 |
| CAF99 | 2016 | 03/2014 - 11/2016 | LS8 | 2019 | 02/2019 - 12/2019 | S-2 |
| CAF05 | 2017 | 12/2014 - 01/2018 | LS8 | 2019 2000 | 02/2019 - 11/2019 11/1999 - 01/2003 | S-2 LS7 |
| CAR01 | 2017 | 05/2016 - 12/2017 | S-2 | 2000 | 02/1999 - 11/2004 | SP4, LS7 |
| EAF04 | 2017 | 04/2016 - 10/2017 | S-2 | 2000 | 07/1999 - 06/2002 | LS7 |
| PAC01 | 2016 | 12/2015 - 11/2016 | S-2 | 2000 2005 2010 | 04/2001 - 11/2002 04/2003 - 12/2007 01/2008 - 10/2012 | SP4, SP5 SP5 SP5, SP6 |
| SAF21 | 2017 | 06/2016 - 11/2017 | S-2 | 2000 | 10/1999 - 12/2002 | LS7 |
| WAF04 | 2017 | 11/2016 - 03/2018 | S-2 | 2000 | 09/1998 - 06/2003 | SP4, SP5 |

*S-2: Sentinel 2; LS7: Landsat 7; LS8: Landsat 8; SP4: SPOT 4; SP5: SPOT 5; SP6: SPOT 6.

**The Discussion Section mentions the direct relationship between population growth and pressure on land but the authors did not present any analysis with empirical data from the pilot countries to that effect in the text. It will be useful to support this claim with data associated with the pilot countries**

We added text and updated the paragraph with additional information as follows:

"The demands of social and economic growth call for additional land, typically at the expense of previously untouched areas. Areas under protection (i.e. national parks) that remain well-preserved (see Figs. 5, 6 and 7) are often in close proximity to regions under excessive pressure. In particular, transboundary areas – such as the mapped W-Arly-Pendjari Complex protected area (WAPOK) – highlight often strong spatial heterogeneity in anthropogenic pressure between

the different countries (Bühne et al., 2017). Such areas need very accurate monitoring and base maps, as provided through this work, especially as areas shared between and/or among countries are frequently not mapped with a common legend, if mapped at all."

**I noticed a number of typos and grammatical forms that have to be revised before the final publication**
The entire manuscript was edited for typos, grammar and typographical/formatting transgressions.